# Bioavailability of Dietary Zinc Sources and Their Effect on Mineral and Antioxidant Status in Lambs

**Ľubomíra Grešáková \*** , **Katarína Tokarčíková and Klaudia Čobanová**

Institute of Animal Physiology, Centre of Biosciences of the Slovak Academy of Sciences, Šoltésovej 4,
040 01 Košice, Slovakia; tokarcikova@saske.sk (K.T.); boldik@saske.sk (K.Č.)
\* Correspondence: gresakl@saske.sk; Tel.: +421-557922970

**Abstract:** This study investigated the relative bioavailability (RBV) of zinc from different sources used as feed additives in ruminant nutrition based on Zn concentration and the activity of Zn-dependent enzymes in lamb tissues. Thirty-two male lambs of Improved Valachian breed (three months old) were randomly assigned to four dietary treatments. For 120 days, the lambs were fed either the total mix ration (TMR) providing 29.6 mg Zn/kg or the TMR supplemented with either zinc sulphate ($ZnSO_4$), zinc chelate of glycine hydrate (ZnGly), or zinc chelate of protein hydrolysate (ZnProt). The supplemented diets contained a total of 80 mg Zn/kg. Supplementation with $ZnSO_4$ increased Zn concentration in the liver, while the highest Zn uptake was in the kidneys of lambs fed the ZnProt diet. The ZnGly supplemented diet elevated the activity of the Cu/Zn-dependent enzyme superoxide dismutase (Cu/Zn SOD) in the liver. Regardless of Zn source, Zn supplementation resulted in increased total antioxidant status (TAS) in the pancreas. The estimated RBV of Zn based on linear regression slope ratios did not differ among the Zn sources. Our results indicate similar availability of Zn from organic dietary sources as from commonly used zinc sulphate; however, their effects on mineral and antioxidant status may differ slightly in growing lambs.

**Keywords:** feed additives; trace minerals; bioavailability; lambs; antioxidant enzymes





## 1. Introduction

Definitions for mineral bioavailability vary because of the complexity of the issue. From a nutritional point of view, the bioavailability of minerals can be defined as the proportion of ingested trace elements utilized for their specific physiological and biochemical functions at the site of action [1]. Mineral bioavailability includes gastrointestinal digestion, absorption, metabolism, tissue distribution, and bioactivity, but it depends on many factors such as feed composition, co-ingested compounds, mineral dosage and chemical form, and the mineral status of animals [2]. Moreover, due to the substantial impacts of homeostatic mechanisms on the plasma and tissue concentrations of minerals, it is difficult to correctly estimate mineral bioavailability based on the above definition.

Zinc as an essential integral part of many important enzyme systems is involved in keeping the health and performance of ruminants at their maximum [3]. Dietary zinc deficiency could lead to impaired growth, reproduction, and immune dysfunction with increased susceptibility to infections in growing animals [3–5]. Although Zn requirements for growing lambs and calves are recommended at 33 mg Zn/kg of dry matter (DM) in a complete diet, young ruminants require additional Zn to support accretion of body protein during periods of rapid growth, and improvement in growth performance, health and reproduction, mineral and antioxidant status, and immunity in ruminants has been reported after Zn supplementation [3,6–9]. In addition, Zn supplementation with Zn nanoparticles can offer an effective approach to maintaining the high production and health of ruminants due to the improvement in animal reproductive efficiency, immunomodulatory properties, and enhancing the microbial biomass production, while it can also help to reduce methane emission in livestock [10].

The methods of assessment of Zn bioavailability include Zn absorption and retention, plasma or tissue Zn concentrations, animal growth, the biologically active form of Zn, and prevention of signs of Zn deficiency [11]. Intake of Zn from organic sources has been found to increase nutrient digestibility and Zn absorption and retention in lambs compared with inorganic Zn supplementation [7,12,13], but the results of other studies focusing on zinc bioavailability in ruminants are contradictory [14–18]. Interaction of Zn with other metal ions in the diet is a dietary factor that can modify Zn bioavailability. Complex interactions among minerals within the digestive tract can affect the absorption of zinc as well as the feed's effectiveness in promoting the health and productivity of ruminants, but organic Zn chelates might permit Zn and other minerals in the diet to circumvent factors that inhibit absorption of their inorganic forms [17,19]. Therefore, in this study, we decided to estimate the relative bioavailability of different zinc sources used as feed additives in ruminant nutrition based on Zn concentrations as well as the activity of Zn-dependent enzymes in lamb tissues and to investigate the effect of organic Zn sources on the mineral and antioxidant status in lambs.

## 2. Materials and Methods

All experimental protocols involving animals were performed in accordance with the Guiding Principles for the Care and Use of Research Animals and Animal Research: Reporting In Vivo Experiments (ARRIVE guidelines). All methods and procedures reported herein were carried out in line with European Union Directive 2010/63/EU for animal experiments, and the experimental protocol was approved by the Ethical Committee of the Institute of Animal Physiology of the Slovak Academy of Sciences and by the State Veterinary and Food Office (Ro-4160/13-221).

### 2.1. Animals, Diets, and Experimental Design

Thirty-two castrated male lambs (Improved Valachian breed) aged three months were selected from the flock of a commercial farm (Olšavica–Brutovce, Slovakia) with an initial mean body weight 19.7 $\pm$ 3.2 kg. The lambs were housed at the Slovak Academy of Sciences Institute of Animal Physiology Research Centre. They were allocated to one of four dietary treatments ($n$ = 8) in a completely randomized design and fed an unsupplemented total mix ration (TMR) consisting of grass hay and feed concentrate (600 and 300 g/day, respectively) in the adaptation period lasting four weeks. During the experimental period, lasting 120 days, at four months of age (mean BW 24.7 $\pm$ 2.97 kg), the lambs were fed the TMR consisting of grass hay and feed concentrate (800 and 350 g/day, respectively) providing a total of 29.6 mg Zn/kg for the control lambs (C, control). The concentrates for the experimental dietary treatments were supplemented either with zinc sulphate as an inorganic Zn source (ZnSO$_4$; ZnSO$_4$·H$_2$O, Sigma–Aldrich, St. Louis, MO, USA), or organic sources of Zn, zinc chelate of glycine hydrate (ZnGly; Glycinoplex-Zn 26%, Phytobiotics Futterzusatzstoffe GmbH, Eltville, Germany) and zinc chelate of protein hydrolysate (ZnProt; Bioplex®-Zn 15%, Alltech Inc., Nicholasville, KY, USA). Commercially produced Zn organic chelates (ZnGly, ZnProt) are more stable complexes produced by reacting Zn with amino acid hydrate or hydrolyzed protein protecting Zn in the gut from the adverse effects of other feed compounds and minerals and make it more absorbable and available to lambs. The TMRs for all dietary treatments were formulated to meet National Research Council (NRC, 2007) requirements [20], and all supplemented diets (TMRs for the supplemented dietary treatments) were prepared so as to contain a total of 80 mg Zn/kg of a complete diet (as-fed basis). The TMR composition is shown in Table 1.

**Table 1.** Ingredients and chemical composition of the total mix ration (TMR) [1] on a dry matter (DM) basis.

| Ingredient | g/kg of DM |
| --- | --- |
| Grass hay | 709 |
| Barley ground | 165 |
| Wheat bran | 81 |
| Soybean meal | 41 |
| Limestone | 4 |
| Chemical composition (analyzed) | |
| Dry matter (g/kg) | 891 |
| Crude protein | 113 |
| Acid detergent fiber | 230 |
| Neutral detergent fiber | 357 |
| Ash | 66 |
| Mn, mg/kg of DM | 34 |
| Zn, mg/kg of DM | 33 |
| Cu, mg/kg of DM | 8.5 |

[1] Common TMR fed to all lambs during the trial. The control received only the TMR, other dietary treatments received the TMR consisting of grass hay and concentrate supplemented with different Zn sources.

During the adaptation period, the lambs were housed in floor pens with bedding ($n = 2$ lambs/pen, $1.64 \times 1.25$ m) equipped with an automatic water supplier and plastic feeders. At the beginning of the experimental feeding period, each lamb was housed individually in a pen and fed the TMR consisting of grass hay and concentrate (800 and 350 g/day, respectively). The feed concentrate contained ground barley, wheat bran, and soybean meal as the main ingredients. The addition of the Zn sources to ground concentrate was analytically confirmed ($n = 8$). The analyzed total Zn concentrations in the TMRs of the supplemented experimental treatments were 78.11 mg Zn/kg for $ZnSO_4$, 79.79 mg Zn/kg for the ZnGly, and 77.49 mg Zn/kg for ZnProt treatments. Throughout the experiment, the animals were fed twice a day and had free access to fresh potable water. Once a week each lamb was offered a trace mineral lick without Zn composed of Ca 16.2, Na 316, Mg 32, Cu 0.5, Co 0.06, I 0.02, and Se 0.01 (g/kg). Feed intake from the daily ration was evaluated separately for each lamb, and the body weights were recorded monthly.

*2.2. Sample Collection and Analysis*

The total amount of feed refused by each lamb was collected and recorded daily at 8:00 a.m. All refused feed samples were dried and ground to pass through a 1 mm screen and stored for subsequent analysis.

Jugular blood samples were collected into acid-washed heparinized tubes from each lamb once a month during the experimental period. Blood samples were centrifuged at $1180 \times g$ for 10 min at 4 °C, and plasma was removed for mineral analysis. Fresh blood samples were immediately used for the determination of superoxide dismutase activity in erythrocytes. At the end of the experiment, six lambs from each treatment group were slaughtered, and tissue samples were collected. Tissue samples were collected from the left lobe of the liver, the left kidney cortex, apex of the heart and the pancreas as well as from the skeletal muscles *m. longissimus dorsi* and *m. psoas major*. Sampling was conducted from identical locations of each relevant tissue using a ceramic knife. All tissues and plasma samples were stored in acid-washed plastic vials or tubes at −80 °C until further analysis.

Dry matter from feed and tissue samples was determined according to the Association of Official Analytical Chemists (AOAC) method [21] by drying samples to a constant weight at 105 °C. After drying the samples of diet ingredients at 60 °C for 48 h, the samples were analyzed for neutral detergent fiber (NDF) and acid detergent fiber (ADF) using the method proposed by Van Soest et al. [22]. NDF was assayed without heat-stable amylase and expressed inclusive of residual ash. ADF was also expressed inclusive of residual ash. ADF and NDF concentrations were determined using the ANKOM2000 Fibre Analyzer

(Ankom Technology, Macedon, NY, USA). Standard methods [21] were used to analyze ash (No. 942.05), nitrogen (No. 968.06), and crude protein (No. 990.03) using the Kjeldahl method after acid digestion.

### 2.3. Mineral Analysis

Trace mineral concentrations in the feed components, plasma, and tissue samples were analyzed using a double-beam atomic absorption spectrophotometer (AA-7000 Series, Shimadzu Co., Kyoto, Japan) with a graphite furnace (GFA-7000, Shimadzu Co., Kyoto, Japan). The microwave-assisted digestion method using closed pressure vessels (MWS 4 Speedwave, Berghof Co., Eningen, Germany) was used for decomposition of all samples in two replicates. All samples, except plasma, were wet acid-digested with a mixture of concentrated nitric acid and hydrogen peroxide. Mineral concentration of Mn in muscles was determined using a graphite furnace atomic absorption spectrophotometer with deuterium background correction and pyrolytic-coated graphite tubes [23]. The certificate reference materials of bovine liver BCR-185, bovine muscle ERM-BB184, and pig kidney ERM-BB186 from the Institute for Reference Materials and Measurements (IRMM, Geel, Belgium) and ClinCheck control of lyophilized human blood (Recipe, Munich, Germany) were included in each analysis to verify instrument accuracy. Mineral concentrations in all samples except plasma were expressed as mg/kg of DM and in plasma as mg/L.

### 2.4. Enzyme Analysis

All antioxidant assays of plasma and animal tissues are described in more detail in the study by Čobanova [14]. The activity of superoxide dismutase (SOD, EC 1.15.1.1) in the erythrocytes was measured using the RANSOD kit (Randox, Crumlin, UK). The activity of SOD was expressed in U/g Hb for the specific activity. Total SOD and Cu/Zn superoxide dismutase (Cu/Zn SOD) activity in the liver, pancreas, and kidney cortex of each lamb was measured by means of spectrophotometric assay according to the procedure proposed by Marklund and Marklund [24], whereby one unit of enzyme activity was defined as the amount of enzyme required to inhibit pyrogallol autoxidation by 50%.

The activity of glutathione peroxidase (GPx, EC 1.11.1.9) in blood and hemoglobin (Hb) content were analyzed using commercial kits from Randox, Crumlin, UK, and GPx activity in lamb tissues was assayed by measuring the oxidation of nicotinamide adenine dinucleotide phosphate (NADPH) according to Paglia and Valentine [25]. The assays were carried out at 25 °C, and hydrogen peroxide was used as the substrate. The enzyme activities of SOD and GPx are expressed as units of enzyme per milligram and per gram of tissue protein, respectively.

The total antioxidant status (TAS) of the blood plasma and lamb tissues was determined by means of the ferric-reducing antioxidant power (FRAP) assay using the modified method of Benzie and Strain [26]. TAS is expressed as μmol $Fe^{2+}$ per gram of tissue protein or mmol $Fe^{2+}$ per liter of plasma.

Lipid oxidation in the serum and tissues was estimated using the modified fluorometric TBARS method according to Jo and Ahn [27]. Lipid peroxidation was expressed as μmol of malondialdehyde (MDA) formed per L of plasma or nmol MDA per gram of tissue protein. Total protein in the tissue homogenates was assayed using the Bradford method [28].

The total concentration of thiol groups (TSH) was found spectrophotometrically based on the reaction of 5,5′-dithio-bis (2-nitrobenzoic acid) with protein thiol groups [29]. The concentrations of the TSH groups are expressed in mmol per liter of plasma or μmol per g of tissue.

Metallothionein 1 (MT1) plasma concentration was determined with the colorimetric method using commercial ELISA kits (Sheep Zn-MT ELISA kit, NeoScientific, London, UK) following the manufacturer's instructions. The optical density of the samples was measured using a microplate reader (Apollo 11 LB913, Berthold Technologies GmHB & Co., KG, Bad Wildbad, Germany).

### 2.5. Statistical Analysis

All statistical analyses were performed using the GraphPad Prism software (GraphPad Prism version 9.0, GraphPad Software, San Diego, CA, USA), and the differences in plasma mineral concentrations between the dietary treatments after 30, 60, and 90 days of Zn supplementation were evaluated using two-way repeated measures ANOVA with the post hoc Tukey's multiple comparisons test (Supplementary Materials, Table S1). Other parameters were analyzed using one-way ANOVA with the post hoc Tukey's multiple comparison test using dietary treatment as the source of variation and lamb as the experimental unit. Differences between the mean values of the different dietary treatments were considered statistically significant at $p < 0.05$. The values in the tables are the means and the pooled standard errors of the mean (SEM).

Relative bioavailability was estimated based on a single dietary Zn concentration from Zn sources, using zinc sulphate as the standard source, by means of slope ratio comparisons from linear regression [30]. Regressions were calculated based on supplemental Zn level as the independent variable, and differences between Zn sources were determined based on the differences in their respective regression coefficients.

## 3. Results

### 3.1. Growth Performance

During the whole experimental period, no form of feed supplementation with Zn from any source influenced either feed intake or body weight/gain compared to unsupplemented control lambs (Table 2).

**Table 2.** Growth performance of lambs for the 120 day feeding period differing in Zn source (80 mg Zn/kg). ADFI: average daily feed intake; ADG: average daily gain.

| Parameters | Dietary Treatments [1] | | | | SEM | *p*-Value [2] |
| --- | --- | --- | --- | --- | --- | --- |
| | **C** | **ZnSO₄** | **ZnGly** | **ZnProt** | | |
| Initial body weight, kg | 24.89 | 24.76 | 24.39 | 24.78 | 0.526 | 0.9981 |
| Final body weight, kg | 31.46 | 30.96 | 31.29 | 31.11 | 0.521 | 0.9892 |
| ADFI, g/d | 1059 | 1059 | 1081 | 1059 | 12.49 | 0.9102 |
| ADG, g/d | 54.82 | 51.64 | 57.60 | 53.74 | 2.484 | 0.8700 |
| Feed/gain | 20.32 | 20.83 | 20.47 | 20.97 | 0.831 | 0.9931 |

[1] C: basal diet, ZnSO₄: zinc sulphate, ZnGly: zinc chelate of glycine hydrate, and ZnProt: zinc chelate of protein hydrolysate. [2] Values ($n = 8$/treatment) are presented as the least squares means.

### 3.2. Mineral Status

The 120 day period of feed supplementation with organic Zn proteinate (ZnProt) increased the plasma Zn concentration compared to the control treatment ($p = 0.028$, Table 3), while the MT concentration in plasma was reduced during the ZnProt treatment ($p < 0.05$). Plasma Cu concentration increased with ZnSO₄ treatment compared to the control and ZnProt treatments ($p < 0.01$) and, therefore, the highest Zn/Cu ratio was observed in the plasma of lambs fed the diet enriched with ZnProt ($p < 0.01$). Plasma concentrations of zinc did not differ between the dietary treatments after 30, 60, and 90 days of Zn supplementation; however, the effects of time and treatment were detected (Supplementary Materials, Table S1). Concentrations of Cu and Fe in plasma were not affected by dietary treatments in all collection periods (i.e., 30, 60, and 90 days of Zn supplementation), but the time effect was detected in both minerals (Table S1).

Feed supplementation with Zn sulphate increased Zn concentration in liver tissue ($p < 0.05$) compared to the control treatment (Table 4). The diet with ZnProt significantly increased Zn concentrations in the kidney cortex compared to other treatments ($p < 0.01$). Zn concentrations in other tissues (muscles, spleen, pancreas, heart, rib bone) were not affected by the dietary treatment.

**Table 3.** Activity of antioxidant enzymes (i.e., SOD and GPx) in blood, plasma mineral concentration, and other plasma parameters of lambs fed diets (80 mg Zn/kg) differing in Zn source after the 120 day feeding period.

| Parameters | Dietary Treatments [1] | | | | SEM | *p*-Value [2] |
|---|---|---|---|---|---|---|
| | **C** | **ZnSO₄** | **ZnGly** | **ZnProt** | | |
| SOD, U/g Hb | 2221 [ab] | 2620 [a] | 1561 [ab] | 1244 [b] | 329 | 0.0402 |
| GPx, U/g Hb | 224 [ab] | 281 [a] | 263 [ab] | 175 [b] | 14.5 | 0.0391 |
| Zn, mg/L | 0.917 [a] | 0.947 [ab] | 0.994 [ab] | 1.15 [b] | 0.029 | 0.0359 |
| Cu, mg/L | 0.535 [a] | 0.773 [b] | 0.703 [ab] | 0.517 [a] | 0.032 | 0.0019 |
| Zn/Cu ratio | 1.09 [ab] | 0.79 [b] | 0.97 [bc] | 1.28 [c] | 0.05 | 0.0024 |
| Fe, mg/L | 2.45 | 2.53 | 2.35 | 2.65 | 0.06 | 0.4136 |
| MT, μg/L | 2.28 [a] | 2.20 [ab] | 2.06 [ab] | 1.95 [b] | 0.03 | 0.0392 |
| ALB, g/L | 34.3 | 33.2 | 34.9 | 35.1 | 0.33 | 0.1474 |
| TSH, mmol/L | 0.389 | 0.412 | 0.393 | 0.447 | 0.01 | 0.1919 |
| TAS, μmol/L | 0.396 | 0.353 | 0.362 | 0.375 | 0.01 | 0.0948 |
| MDA, μmol/L | 0.160 | 0.146 | 0.175 | 0.163 | 0.01 | 0.4099 |

SOD: superoxide dismutase, GPx: glutathione peroxidase, MT: metallothionein, ALB: albumin, TSH: total thiol groups, TAS: total antioxidant status, and MDA: malondialdehyde. [1] C: basal diet, ZnSO₄: zinc sulphate, ZnGly: zinc chelate of glycine hydrate, and ZnProt: zinc chelate of protein hydrolysate. [2] Values (*n* = 6/treatment) are presented as the least squares means. Means within lines with different superscript letters are significantly different (*p* < 0.05).

**Table 4.** Tissue Zn concentrations in lambs fed diets differing in Zn source (80 mg Mn/kg) for the 120 day feeding period.

| Tissue, mg/kg of DM | Dietary Treatments [1] | | | | SEM | *p*-Value [2] |
|---|---|---|---|---|---|---|
| | **C** | **ZnSO₄** | **ZnGly** | **ZnProt** | | |
| Liver | 118.4 [a] | 134.7 [b] | 130.3 [ab] | 126.4 [ab] | 1.99 | 0.0354 |
| Kidney | 109.9 [a] | 110.6 [a] | 107.8 [a] | 120.2 [b] | 1.41 | 0.0038 |
| Muscle | | | | | | |
| *Longissimus dorsi* | 109.8 | 108.8 | 103.8 | 107.7 | 2.04 | 0.7508 |
| *Psoas major* | 101.3 | 105.9 | 106.6 | 109.3 | 2.36 | 0.6981 |
| Spleen | 97.64 | 97.83 | 101.3 | 101.6 | 1.12 | 0.5772 |
| Pancreas | 68.9 | 66.9 | 68.5 | 70.1 | 0.62 | 0.5322 |
| Heart | 68.11 | 66.57 | 68.96 | 66.86 | 0.76 | 0.7805 |
| Rib bone, mg/kg ash | 140.9 | 132.1 | 131.3 | 146.2 | 2.44 | 0.1764 |

[1] C: basal diet, ZnSO₄: zinc sulphate, ZnGly: zinc chelate of glycine hydrate, and ZnProt: zinc chelate of protein hydrolysate. [2] Values (*n* = 6/treatment) are presented as the least squares means. Means within lines with different superscript letters are significantly different (*p* < 0.05).

The tissue concentrations of Mn, Fe, and Cu in liver, kidney, muscles, spleen, pancreas, heart and rib bone are presented in Table 5. Increased Cu concentrations in the pancreas and decreased Mn concentrations in the heart muscle were observed in the lambs fed diets supplemented with both organic Zn sources (*p* < 0.01 and *p* < 0.05, respectively). The ZnProt-enriched diet decreased liver Cu concentrations compared to the control lambs (*p* < 0.05). Cu concentrations increased in the kidney cortex of lambs given ZnSO₄ and ZnGly treatments compared to the control animals (*p* < 0.001). Feed supplementation with ZnGly decreased Fe concentrations in kidney and muscles (*m. psoas major*) compared to ZnProt treatment (*p* < 0.05) and decreased muscle Mn (*m. longissimus dorsi*) compared with ZnSO₄ but increased Mn concentrations in the pancreas in comparison with the control and ZnProt treatments. Intake of the diets supplemented with Zn from both organic sources decreased Fe concentrations in the heart compared to the control and ZnSO₄ treatment (*p* < 0.01).

**Table 5.** Tissue concentrations of Cu, Fe, and Mn in lambs fed diets differing in Zn source (80 mg Mn/kg) for the 120 day feeding period.

| Tissue, mg/kg of DM | Dietary Treatments [1] | | | | SEM | *p*-Value [2] |
| --- | --- | --- | --- | --- | --- | --- |
| | C | ZnSO$_4$ | ZnGly | ZnProt | | |
| **Copper** | | | | | | |
| Liver | 302.2 [a] | 240.0 [ab] | 252.7 [ab] | 221.1 [b] | 10.4 | 0.0353 |
| Kidney | 20.13 [a] | 21.32 [b] | 20.52 [ab] | 21.40 [b] | 0.17 | 0.0011 |
| Pancreas | 4.384 [a] | 4.701 [a] | 5.232 [b] | 5.378 [b] | 0.12 | 0.0081 |
| Heart | 14.56 | 14.01 | 13.81 | 12.47 | 0.29 | 0.1384 |
| Spleen | 4.233 | 4.142 | 4.207 | 4.147 | 0.12 | 0.9953 |
| Muscle | | | | | | |
| *Longissimus dorsi* | 2.853 | 2.611 | 2.899 | 2.605 | 0.21 | 0.9493 |
| *Psoas major* | 1.278 | 1.187 | 1.260 | 1.281 | 0.07 | 0.9705 |
| **Iron** | | | | | | |
| Liver | 230.5 | 188.0 | 232.8 | 230.0 | 13.4 | 0.6152 |
| Kidney | 204.2 [a] | 199.1 [ab] | 172.5 [b] | 207.8 [a] | 5.00 | 0.0164 |
| Pancreas | 73.47 | 74.62 | 75.00 | 87.15 | 2.15 | 0.2443 |
| Heart | 150.4 [a] | 151.7 [a] | 135.2 [b] | 135.2 [b] | 2.13 | 0.0042 |
| Spleen | 1364 | 975.4 | 1037 | 1531 | 114.6 | 0.4184 |
| Muscle | | | | | | |
| *Longissimus dorsi* | 63.09 | 63.12 | 56.06 | 69.54 | 2.13 | 0.2997 |
| *Psoas major* | 56.07 [ab] | 60.31 [ab] | 51.84 [a] | 62.02 [b] | 1.23 | 0.0300 |
| **Manganese** | | | | | | |
| Liver | 8.733 | 9.124 | 9.579 | 9.990 | 0.29 | 0.5105 |
| Kidney | 5.087 | 5.496 | 5.320 | 5.077 | 0.11 | 0.4678 |
| Pancreas | 7.134 [a] | 8.021 [ab] | 8.440 [b] | 8.121 [a] | 0.16 | 0.0126 |
| Heart | 1.340 [ab] | 1.408 [a] | 1.300 [b] | 1.133 [b] | 0.04 | 0.0623 |
| Spleen | 1.214 | 1.195 | 1.439 | 1.393 | 0.04 | 0.1432 |
| Muscle | | | | | | |
| *Longissimus dorsi* | 0.321 [ab] | 0.348 [a] | 0.237 [b] | 0.327 [ab] | 0.02 | 0.0450 |
| *Psoas major* | 0.334 | 0.368 | 0.275 | 0.278 | 0.02 | 0.1004 |

[1] C: basal diet, ZnSO$_4$: zinc sulphate, ZnGly: zinc chelate of glycine hydrate, and ZnProt: zinc chelate of protein hydrolysate. [2] Values (*n* = 6/treatment) are presented as the least squares means. Means within lines with different superscript letters are significantly different (*p* < 0.05).

*3.3. Antioxidant Status*

Activity of both antioxidant enzymes SOD and GPx decreased in the erythrocytes of lambs on the diet supplemented with ZnProt compared to ZnSO$_4$ treatment only (both *p* < 0.05). Other parameters of plasma antioxidant status (ALB, TAS, TSH, and MDA) were not affected by dietary treatment (Table 3).

The highest activity of total SOD as well as of Cu/Zn SOD was observed in liver tissue of lambs fed the ZnGly-enriched diet compared to all other lambs (*p* < 0.0001, Table 6). SOD activity in other tissues was not affected by the dietary treatment. Regardless of the source, Zn supplementation increased total antioxidant status (TAS) in the pancreas of all treated lambs (*p* < 0.0001). Other antioxidant parameters, such as GPx activity or concentrations of MDA, TSH, and NPSH in liver, kidney and pancreas tissues, were not affected by Zn feed supplementation.

**Table 6.** Antioxidant parameters in tissues of lambs fed diets (80 mg Zn/kg) differing in Zn source [1].

| Enzyme Activity | Dietary Treatments [1] | | | | SEM | *p*-Value [2] |
|---|---|---|---|---|---|---|
| | C | ZnSO$_4$ | ZnGly | ZnProt | | |
| **Liver** | | | | | | |
| SOD, U/mg protein | 90.8 [a] | 123 [b] | 161 [c] | 108 [ab] | 6.60 | 0.0001 |
| Cu/Zn SOD, U/mg protein | 78.7 [a] | 107 [ab] | 130 [b] | 95.4 [a] | 5.45 | 0.0021 |
| GPx, U/g protein | 19.6 | 21.6 | 19.7 | 19.6 | 0.55 | 0.5282 |
| MDA, nmol/g protein | 157 | 165.5 | 142.1 | 161.8 | 6.53 | 0.6374 |
| TAS, µmol/g protein | 38.7 | 37.37 | 37.58 | 39.96 | 1.20 | 0.8805 |
| TSH, µmol/g tissue | 15.5 | 16.0 | 15.3 | 15.3 | 0.24 | 0.8028 |
| NPSH, µmol/g tissue | 5.20 | 4.75 | 4.50 | 4.73 | 0.10 | 0.1481 |
| **Kidney cortex** | | | | | | |
| SOD, U/mg protein | 55.9 | 47.4 | 50.0 | 46.9 | 2.17 | 0.4627 |
| Cu/Zn SOD, U/mg protein | 44.8 | 35.6 | 37.8 | 37.8 | 2.03 | 0.4218 |
| GPx, U/g protein | 21.5 | 21.9 | 21.5 | 21.3 | 0.62 | 0.9907 |
| MDA, nmol/g protein | 74.2 | 75.2 | 69.6 | 80.4 | 1.95 | 0.3056 |
| TAS, µmol/g protein | 21.5 | 21.4 | 21.5 | 21.9 | 0.51 | 0.9896 |
| TSH, µmol/g tissue | 8.31 | 8.54 | 8.31 | 9.32 | 0.16 | 0.1426 |
| NPSH, µmol/g tissue | 2.44 | 2.70 | 2.43 | 2.71 | 0.05 | 0.1242 |
| **Pancreas** | | | | | | |
| SOD, U/mg protein | 5.38 | 5.29 | 5.78 | 4.20 | 0.36 | 0.4847 |
| Cu/Zn SOD, U/mg protein | 5.12 | 4.93 | 5.51 | 4.04 | 0.35 | 0.5142 |
| GPx, U/g protein | 17.9 | 14.7 | 19.8 | 17.3 | 1.10 | 0.4590 |
| MDA, nmol/g protein | 69.6 | 72.9 | 69.6 | 71.7 | 3.16 | 0.9801 |
| TAS, µmol/g protein | 3.31 [a] | 6.68 [b] | 9.08 [b] | 7.40 [b] | 0.55 | 0.0001 |
| TSH, µmol/g tissue | 8.70 | 9.05 | 10.11 | 8.93 | 0.26 | 0.3787 |
| NPSH, µmol/g tissue | 1.92 | 1.73 | 1.98 | 1.71 | 0.06 | 0.3463 |

[1] C: basal diet, ZnSO$_4$: zinc sulphate, ZnGly: zinc chelate of glycine hydrate, and ZnPro: zinc chelate of protein hydrolysate. [2] Values (*n* = 6/treatment) are presented as the least squares means. Means within lines with different superscript letters are significantly different (*p* < 0.05).

### 3.4. Bioavailability

Relative bioavailability (RVB) of Zn sources was estimated based on the linear regression slope ratios of Zn concentrations and the activity of the Zn-dependent enzyme Cu/Zn SOD in lamb tissues regressed on the basis of Zn supplemental levels (Table 7). ZnSO$_4$ used as the Zn standard was assigned a value of 100%. The slopes and estimated RBV values based on Zn concentrations in plasma, liver, and kidney as well as liver activity of Cu/Zn SOD did not differ among the Zn feed additives (*p* > 0.05). The best parameter for the assessment of RVB was Cu/Zn SOD activity in the liver ($R^2$ = 0.44), and based on this parameter, RBV tended to be higher in lambs fed ZnGly (*p* < 0.085).

**Table 7.** Estimated relative bioavailability (RBV) of Zn sources based on linear regression slope ratios of Zn concentrations in tissues and liver activity of Cu/Zn SOD on Zn supplementation level (mg).

| Dependent Variable | Zinc Source | Regression Coefficient | | RBV, % | *p*-Value |
|---|---|---|---|---|---|
| | | Slope | SE | | |
| Plasma Zn, mg/L [a] | Zn sulphate | 0.000405 | 0.000576 | 100 | |
| | Zn glycinate | 0.001357 | 0.000839 | 335 | 0.6154 |
| | Zn proteinate | 0.001029 | 0.000641 | 254 | |
| Liver Zn, mg/kg DM [b] | Zn sulphate | 0.2335 | 0.0841 | 100 | |
| | Zn glycinate | 0.1903 | 0.1028 | 81.5 | 0.7136 |
| | Zn proteinate | 0.1309 | 0.0679 | 56.1 | |
| Kidney Zn, mg/kg DM [c] | Zn sulphate | 0.1220 | 0.0726 | 100 | |
| | Zn glycinate | 0.1046 | 0.0711 | 85.7 | 0.8926 |
| | Zn proteinate | 0.1538 | 0.0776 | 126 | |
| Liver Cu/Zn SOD, U/mg protein [d] | Zn sulphate | 0.4089 | 0.1450 | 100 | |
| | Zn glycinate | 0.7338 | 0.1840 | 179 | 0.085 |
| | Zn proteinate | 0.2378 | 0.1272 | 58.2 | |

[a] Intercept = 0.57, $R^2$ = 0.16, and SD = 0.086; [b] Intercept = 118, $R^2$ = 0.33, and SD = 11.9; [c] Intercept = 106, $R^2$ = 0.23, and SD = 9.72; [d] Intercept = 78.7, $R^2$ = 0.44, and SD = 24.6.

## 4. Discussion

Twelve-week feed supplementation with 80 mg of Zn/kg did not affect plasma Zn concentrations measured once a month compared to control. However, after the 120 day feeding period with differing Zn dietary sources, the plasma Zn concentration increased in lambs fed the diet supplemented with ZnProt. In many studies, long-term supplementation of the basal diet with a low Zn concentration range (10–30 mg of Zn/kg) as well as Zn dietary source affected neither plasma nor tissue Zn concentrations in ruminants [6,12,31–33]. We decided to supplement the TMR with close to the proposed total maximum contents of Zn for ruminants in the EU [34] to contain a total of 80 mg Zn/kg of complete diet from different Zn sources. Zn plasma levels reflected dietary Zn concentrations, when sheep were fed with Zn-marginal or -deficient diets containing only 14–22 mg Zn/kg DM [17,35,36]. Our results indicate that the unsupplemented TMR containing 30 mg of Zn/kg was sufficient to meet the physiological requirements for growing lambs. Because the offered TMR was not Zn-deficient and plasma Zn concentrations were maintained within a relatively narrow value span because of efficient homeostatic regulation [37], differences in plasma Zn levels between the dietary treatments were not significant. Although we found a time effect on Zn plasma concentrations, a long period of Zn supplementation caused changes in Zn absorption and excretion to maintain homeostasis; therefore, no differences in Zn bioavailability were detected.

Although Zn plasma seems to not be a reliable marker for assessment of Zn bioavailability [3,12], and we found positive correlation between Zn concentrations in plasma and pancreas, liver, and kidney. Zn concentrations in liver, kidney, pancreas, and bone are considered as suitable markers for assessment of Zn bioavailability in ruminants [11,31,38]. However, increased Zn concentrations were observed mainly in these tissues in ruminants fed diets supplemented with high Zn concentrations (300–1400 mg Zn/kg), with the greatest Zn deposition in plasma and/or tissues in animals fed diets enriched with organic Zn sources [33,39,40]. Differences between Zn feed additives have not been found in tissue mineral deposition using low or normal Zn supplementation levels [6,31–33,41]. Despite the fact that Zn concentrations in tissues are controlled by homeostatic changes in Zn absorption and/or endogenous excretion [4,37], and we supplemented our lambs' diet with up to 80 mg of Zn/kg, increased Zn concentrations were found in the kidneys of lambs fed the ZnProt diet compared to other supplemented or unsupplemented lambs, and Zn levels were elevated in the liver of lambs receiving $ZnSO_4$ treatment compared to controls but with no differences between Zn sources. Calves receiving $ZnSO_4$ at 20 mg of Zn/kg for 56 or 98 days had higher Zn content in their liver than animals fed ZnProt, but after increasing supplemental Zn levels to 500 mg Zn/kg, greater absorption and retention of Zn from ZnProt were observed, resulting in increased Zn deposition in liver, kidney, and plasma [31]. Our results suggest that feed supplementation with Zn from different sources at 80 mg of Zn/kg of diet resulted in Zn being absorbed and utilized at similar rates, and higher Zn levels in the kidneys of lambs fed the ZnProt diet may indicate some differences in the post-absorptive metabolism of Zn from an organic source [9,16,31].

Zinc as an activator of the enzyme SOD as well as other antioxidant proteins and molecules contributes to the proper functioning of the antioxidant defense system [5,42,43]. Regardless of Zn source, Zn supplementation improves the antioxidant status of ruminants due to the increased antioxidant activity of SOD and glutathione peroxidase (GPx) [7,14,44], while in our study, the total antioxidant status improved in the pancreas and the SOD activity in the liver of all supplemented lambs. Intake of supplemented diets with zinc sulphate and Zn glycinate significantly increased liver activity of total SOD, and the highest Cu/Zn SOD activity was determined in the liver of lambs given the ZnGly diet only. It seems that ZnGly could increase the antioxidant activity of Cu/Zn SOD in the liver, which is supported by the hypothesis of higher Zn availability from ZnGly maintaining the normal enzyme activity in the body; however, ZnGly treatment affected Cu/Zn SOD and/or ALP activity in the liver and serum of monogastric animals [45–47]. Zn supplementation from organic Zn sources increased Cu/Zn SOD activity as well as Zn concentrations in the liver

of ruminants compared to ZnSO$_4$ [31,33,48,49]. Surprisingly, we found remarkably reduced activity in both antioxidant enzymes, SOD and GPx, in the liver and/or blood as well as plasma metallothionein concentration in lambs supplemented with ZnProt; however, the lambs receiving ZnProt treatment had the highest Zn concentration in kidney and plasma after the 120 d feeding period. This could indicate that Zn from ZnProt was not associated with SOD enzyme synthesis and may be metabolized differently from other Zn sources [31].

The most appropriate criteria for estimating the relative bioavailability (RVB) of zinc in mature ruminants is Zn concentration in the liver, kidney, and pancreas or liver metallothionein concentration [3,31,39,50], when the RVB of zinc sources was expressed relative to zinc sulphate as a standard source. We decided to use Zn concentrations in liver, kidney, and pancreas tissue to estimate the RVB of Zn sources; however, the tissue Zn deposition in particular did not differ significantly from the other supplemented treatments. Since the bioavailability of minerals is defined as the proportion of ingested element from feed which is utilized for specific physiological and biochemical functions at the site of action [1,51], we also used the liver activity of Cu/Zn SOD to estimate RVB, as that was where the significant differences between the individual treatments were found. Our results show that the RBV of Zn did not differ among the Zn sources, because there were no significant differences between their linear regression slopes. The best parameter for bioavailability assessment appears to be the activity of Cu/Zn SOD in the liver, as the highest coefficient of determination for this parameter was found there. This is in accord with other studies reporting no differences in Zn utilization and RVB among Zn sources in ruminants [33,39,41,52]. Potentially better bioavailability of organic Zn chelates or complexes in ruminants has been reported in a few studies based on improved gut absorption, tissue retention of Zn, and/or higher activity of Cu/Zn SOD in liver [17,48]. Cao et al. [39] reported that Zn proteinate was a more available source of Zn for lambs than ZnSO$_4$ and other organic Zn sources due to the higher Zn uptake by kidney and pancreas tissues. Although we also recorded higher Zn concentrations in kidney and plasma of lambs fed the ZnProt diet and liver activity of SOD in ZnGly treatment, we assumed similar relative bioavailability of all Zn sources used based on our linear regression slope ratio comparisons.

Potentially greater absorption of Zn from organic sources, slower release of Zn from Zn chelates or complexes and differing post-absorptive metabolism may affect mineral tissue deposition and mineral status in ruminants. Zn, Cu, Mn, and Fe are chemically similar, so interaction of these elements appears to be tissue-specific due to the competition among the minerals sharing the same transporter systems at the level of membrane transport [53,54]. It has been suggested that Zn supplementation stimulates the production of metallothioneins (MTs) in the intestine and other tissues [55]. In our study Zn supplementation affected mineral status in lambs due to the different mineral tissue deposition. Intake of the diets supplemented with Zn from both organic sources resulted in increased Cu deposition in the pancreas and reduced Mn uptake by the heart muscle. ZnProt supplementation reduced plasma and hepatic Cu concentrations, which could indicate sequestration of dietary Cu in intestinal MT induced by feeding with the ZnProt diet. Low intracellular available Cu bound to MT could affect Fe and Mn transport through the Cu-dependent protein hephaestin and a common transporter, ferroportin [56,57]. Unfortunately, we did not measure MT concentrations in the intestine mucosa, so our results cannot clearly indicate binding of Cu to MT. Further investigation is needed to elucidate whether Zn intake from ZnProt can induce a high metallothionein level in the intestinal mucosa and post-absorptive metabolism of ZnProt as well.

## 5. Conclusions

We can conclude that organic Zn sources, zinc chelate of glycine hydrate, and zinc chelate of protein hydrolysate, given adequate levels of dietary Zn, were absorbed and utilized in a similar way as inorganic Zn feed additive zinc sulphate in growing lambs,

and Zn supplementation increased antioxidant status in the pancreas regardless of Zn source. Relative bioavailability of zinc did not differ among the Zn feed additives; however, the effect of Zn from organic sources on mineral deposition and antioxidant status may differ slightly in lambs. Further research is needed in order to elucidate the differences in post-absorptive metabolism of Zn from organic sources.

Similarities in bioavailability of Zn sources indicate that the lambs' Zn requirements were met (33 mg Zn/kg of DM), regardless of Zn supplemental source. However, feed supplementation with Zn from the inorganic and organic sources at 80 mg of Zn/kg of complete feedstuffs might improve the antioxidant status of growing ruminants without any effect on the growth performance.

**Supplementary Materials:** The following are available online at https://www.mdpi.com/article/10.3390/agriculture11111093/s1, Table S1. Trace mineral concentrations in plasma after 30, 60 and 90 days of Zn supplementation (80 mg Zn/kg).

**Author Contributions:** Conceptualization, Ľ.G. and K.Č.; methodology, Ľ.G. and K.Č.; validation, Ľ.G. and K.Č.; formal analysis, Ľ.G., K.T. and K.Č.; investigation, Ľ.G. and K.Č.; data curation, Ľ.G. and K.Č.; writing—original draft preparation, Ľ.G.; writing—review and editing, Ľ.G. and K.Č.; project administration, Ľ.G. and K.Č.; funding acquisition, K.Č. All authors have read and agreed to the published version of the manuscript.

**Funding:** This research was funded the Slovak Research and Development Support Agency APVV (grant number: 17-0297) and by the Slovak Grant Agency VEGA (grant number: 2/0008/21).

**Institutional Review Board Statement:** This study was conducted according to the guidelines of European Union Directive 2010/63/EU for animal experiments and approved by the Ethics Committee of the Institute of Animal Physiology of the Slovak Academy of Sciences and by the State Veterinary and Food Office (Ro-4160/13-221, 2.12.2013).

**Informed Consent Statement:** Not applicable.

**Data Availability Statement:** The datasets used and analyzed in this survey are available from the corresponding authors upon reasonable request.

**Acknowledgments:** We thank Andrew Billingham for providing language support that greatly improved the manuscript. Thanks go to Renata Gerocova for their excellent assistance in data collection.

**Conflicts of Interest:** The authors declare no conflict of interest.

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
