# Peer review of "Bioavailability of Dietary Zinc Sources and Their Effect on Mineral and Antioxidant Status in Lambs"

_agriculture, doi:10.3390/agriculture11111093_

Round 1
Reviewer 1 Report
This article investigates the bioavailability of different dietary Zn sources and their effects on antioxidant status of growing lambs. The aim of the study is good in general. Although, some details and clarifications are required before deciding if this article can be handled further.
Abstract
line 18: explain what is RBV is
Introduction
Information regarding the biological role of Zn in ruminants (lambs or growing animals) is required, as well as a background on animals’ requirements from this trace mineral. In another mean, why Zn should be supplemented specially if the diet meets animals’ requirements with no negative effects on animals’ health and/or performance.
As other minerals were determined in this study, more information about the biological relationship between Zn and selected minerals should be highlighted.
Materials and methods
Line 62: zinc chelate of glycine hydrate (ZnGly) or zinc chelate of protein hydroly-sate (ZnProt). It is not enough to mention only the name of Zn supplement. More details regarding manufacturing company, classification (organic, non-organic), and the concentration of Zn in the commercial product, the difference between ZnGly and ZnProt in terms of the organic moiety and its biological activity are all required data.
Lines 70-78: I could not understand whether the lambs were fed a TMR (as mention in Table 1, grass hay was a part of the basal diet) or concentrate diet and grass hay were provided separately as mentioned latter.
Line 77: complete experimental diet, what do you mean by complete ??
Line 64: what do you mean by complete diet, please use more specific term such as: as fed and/or on dry matter base.
The animals were fed predetermined amounts of diets, how you ensure that you can obtain refuses to estimate the feed intake. Were the amounts offered higher than required by animals at this growing stage?
Line 86: In addition to the intake of the daily ration, the total amount……Please, remove: In addition to the intake of the daily ration
Line 90-91: remove to tissue collection paragraphs, this paragraph specifies blood sampling and related analyses.
Line 101: (60°C, 48 h) what do these conditions refer to??
For abbreviations: eg: SOD, GPx, …….and other please, write the complete name at first appearance in the text then use the abbreviated term.
Statistical analysis: one way anova may be suitable for the analyses performed in tissues (one time after slaughter), however for blood plasma parameters this seems not true. Blood plasma were monthly collected as stated in previous section. So, you have to consider time effect as a fixed factor. Unless, the blood samples were pooled. You can use one way anova. Whatever the fact this should be explained. Also, the authors have to consider this one of the defects in this study as time effect on Zn bioavailability was not considered. This may partially explain why no differences in Zn bioavailability were detected. This the action of homeostasis.
Table 7: please, amend its layout. Rows are not fit each corresponding parameter.
Discussion: is well written, however I am still not satisfied regarding the lack of bioavailability of Zn from organic sources compared to non-organic source. This may relate to the ignorance of time effect.
Conclusion: In this part, the authors should post recommendation about the need of supplementary Zn in general and what are the recommendations regarding the dose and the source of Zn that should be provided for growing lambs.
Reviewer 2 Report
The paper is interesting and, in my opinion, the authors well described the aim and the adopted methodologies. In addition, they well discusses their results.
line 190: please delete Zn. Actually its plasma concetration was affected by dietary treatment (see table 3)
line 237: please replace table 2 with table 3.
Round 2
Reviewer 1 Report
The authors did good work in the manuscript and satisfy most of my comments. Only, the part regarding the diet and its Zn content is still not clear.
In table 1: the BD consists of 709 g grass hay/kg diet and the rest was a mixture of soybean and grains, then the authors stated in line 100 that the BD consisting of grass hay and concentrate was provided at 600 g and 300 g/day respectively. Again, there is a conflict in this part. Do you mean that the BD contains both grass hay and concentrate together and provided with 300 g /animal/day in addition to 600 g/animal/day grass hay???
If so, please use the term BD referring to the concentrate diet only. If no, please rewrite this part.
Also, was Zn concentration analyzed in concentrate diet (BD if I understand correctly) only or whole presented diet (concentrate + grass hay).
I am not sure also, if 11.3% crude protein is enough ratio to growing lambs.
The following recent publication may be useful for your introduction and discussion.
Abdelnour, S.A.; Alagawany, M.; Hashem, N.M.; Farag, M.R.; Alghamdi, E.S.; Hassan, F.U.; Bilal, R.M.; Elnesr, S.S.; Dawood, M.A.O.; Nagadi, S.A.; Elwan, H.A.M.; ALmasoudi, A.G.; Attia, Y.A. Nanominerals: Fabrication Methods, Benefits and Hazards, and Their Applications in Ruminants with Special Reference to Selenium and Zinc Nanoparticles. Animals 2021, 11, 1916. https://doi.org/10.3390/ani11071916
